# Anthropometric indices and cut-off points in the diagnosis of metabolic disorders

**Stanisław Głuszek**[1], **Elzbieta Ciesla**[2], **Martyna Głuszek-Osuch**[2], **Dorota Kozieł**[2], **Wojciech Kiebzak**[2], **Łukasz Wypchło**[1], **Edyta Suliga**[2]*

1 Institute of Medical Sciences, Medical College, Jan Kochanowski University, Kielce, Poland, 2 Institute of Health Sciences, Medical College, Jan Kochanowski University, Kielce, Poland

* edyta.suliga@ujk.edu.pl

## Abstract

### Objective

Identifying metabolic disorders at the earliest phase of their development allows for an early intervention and the prevention of serious consequences of diseases. However, it is difficult to determine which of the anthropometric indices of obesity is the best tool for diagnosing metabolic disorders. The aims of this study were to evaluate the usefulness of selected anthropometric indices and to determine optimal cut-off points for the identification of single metabolic disorders that are components of metabolic syndrome (MetS).

### Design

Cross-sectional study.

### Participants

We analyzed the data of 12,328 participants aged 55.7±5.4 years. All participants were of European descent.

### Primary outcome measure

Four MetS components were included: high glucose concentration, high blood triglyceride concentration, low high-density lipoprotein cholesterol concentration, and elevated blood pressure. The following obesity indices were considered: waist circumference (WC), body mass index (BMI), waist-to-height ratio (WHtR), body fat percentage (%BF), Clínica Universidad de Navarra-body adiposity estimator (CUN-BAE), body roundness index (BRI), and a body shape index (ABSI).

### Results

The following indices had the highest discriminatory power for the identification of at least one MetS component: CUN-BAE, BMI, and WC in men (AUC = 0.734, 0.728, and 0.728, respectively) and WHtR, CUN-BAE, and WC in women (AUC = 0.715, 0.714, and 0.712, respectively) (p<0.001 for all). The other indices were similarly useful, except for the ABSI.

**Data Availability Statement:** All relevant data are within the paper and its Supporting Information files.

**Funding:** The project was supported under the program of the Minister of Science and Higher Education under the name "Regional Initiative of Excellence" in 2019–2022, project number: 024/RID/2018/19, financing amount: 11.999.000,00PLN. The funders had no role in study design, data collection and analysis, decision to publish, or preparation of the manuscript.

**Competing interests:** The authors have declared that no competing interests exist.

## Conclusions

For the BMI, the optimal cut-off point for the identification of metabolic abnormalities was 27.2 kg/m$^2$ for both sexes. For the WC, the optimal cut-off point was of 94 cm for men and 87 cm for women. Prospective studies are needed to identify those indices in which changes in value predict the occurrence of metabolic disorders best.

## Introduction

Overweight and obesity significantly increase the probability of metabolic disorders and chronic diseases like diabetes; cardiovascular diseases, particularly heart failure and coronary heart disease; nonalcoholic fatty liver disease; neoplasms (endometrial, breast post-menopause, prostate, liver, pancreas, colorectum, and kidney); musculoskeletal disorders; and respiratory diseases [1–5]. Overweight or obesity (body mass index [BMI] $\geq$25 kg/m$^2$) characterize 82.5%, 76.4%, and 73.6% of people with Type 2 diabetes, hypertension, and dyslipidemia, respectively [6].

Excess adipose tissue favors the release of free fatty acids (FFAs) from adiposities to the cardiovascular system [1,6]. This leads to an increase in lipid accumulation, including in hepatocytes and skeletal muscle cells. The accumulation of lipids in these cells through the activation of the diacylglycerol-protein kinase C pathway may contribute to the emergence of insulin resistance [7]. Weight gain accompanied by an increase in body fat leads to insulin resistance and to disorders in the expression of various adipokines. This includes classical hormones, such as leptin; inflammatory mediators like tumor necrosis factor-alfa, interleukins -1, -6, and -8, resistin, and chemerin; enzymes; and metabolites [1, 6, 7].

The pathophysiology of obesity-induced dyslipidemia involves, amongst other disorders, a reduced lipolysis of triglyceride (TG)-rich lipoproteins, impairment of the peripheral uptake of FFAs, increased FFA flow from adiposities to the liver and other tissues, overproduction of very low density lipoproteins (VLDL) in the liver, and formation of small dense low density lipoproteins (LDL) [8]. These disorders lead to the development of nonalcoholic fatty liver disease. A decrease in high density lipoprotein (HDL) concentration in obesity involves both an increased uptake of the HDL2 subfraction by adiposities and the increased catabolism of the HDL apolipoprotein A-I [9]. Insulin resistance also contributes to changes in lipid metabolism and the development of atherogenic dyslipidemia [6]. The failure to inhibit the microsomal TG transfer protein and lipoprotein lipase activation observed in insulin resistance leads to hypertriglyceridemia [10].

The most important mechanisms in the development of obesity-induced hypertension includes the physical compression of the kidneys by accumulating fatty cells, which impedes sodium excretion, followed by the activation of the renin-angiotensin-aldosterone system (RAAS). There is also an increase in the activity of the sympathetic nervous system, which is probably caused by elevated leptin secretion, the activation of the melanocortin system in the brain, and resistance to insulin [11, 12]. Obesity is also related to the overproduction of adipokines, which disturb the functions of the endothelium of the blood vessels, vasoconstriction, and vasodilation [12]. Consequences of these disorders include metabolic syndrome (MetS) and cardiovascular diseases.

The early identification of metabolic disorders allows for an early intervention and the prevention of serious consequences. Overweight and obesity indices, calculated based on anthropometric measurements, have found a wide application in the identification of metabolic

disorders [13–18]. The most significant advantages of anthropometric indices include the following: non-invasiveness, low cost, standardized techniques and simplicity of measurements, and the possibility to apply them on a large scale. However, it is still difficult to decide which of the indices is the best tool for the early identification of metabolic disorders. In this study, we analyzed both traditional obesity indicators, which have been used for many decades, and those that have been developed relatively recently. We selected indicators that can be calculated based on 2–3 of the simplest anthropometric measurements and that assess not only weight and height proportions (BMI) but also overall body fat (body fat percentage [%BF], Clínica Universidad de Navarra-body adiposity estimator [CUN-BAE]) and fat distribution (waist circumference [WC], waist-to-height ratio [WHtR]). We also selected indicators that combine several measurements defining the geometry of the human body (body roundness index [BRI], a body shape index [ABSI]) and which were developed using allometric analysis. The usefulness of the BMI for estimating obesity is limited because the values at the high end of the BMI scale can be attributed to either increased fat mass or lean body mass. The commonly used BMI classification excludes people with an increased metabolic risk resulting from high body fat [19]. The CUN-BAE is based on the BMI, but it has the advantage of taking into account age and sex. The percentage of fat calculated using the CUN-BAE showed a stronger correlation with the actual amount of adipose tissue than any other anthropometric fat indicator [19]. Many studies have shown that the results of central obesity measurements have the strongest correlations with metabolic risk factors [20]. The disadvantage of the WC is that taller people have larger circumferences. Also, WC values differ between ethnic groups [21]. However, some studies have found that the use of abdominal obesity measurements does not improve the prediction of metabolic risk factors compared with the BMI [22, 23]. According to Ashwell and Gibson, the WHtR identifies more people with cardiovascular risk than a combination of the WC and the BMI [24]. Other studies indicate that the usefulness of the WHtR differs depending on sex [25]. The ABSI is minimally correlated with height, mass, and the BMI, and can therefore be used to distinguish the independent contributions of WC and BMI to cardiometabolic risk factors [26]. The ABSI had a positive linear relationship with all-cause and cardiovascular mortality in Europeans, while the corresponding relationship with BMI, WC, and WHtR was J-shaped [27]. The ABSI was also useful to identify visceral and sarcopenic obesity in patients with diabetes [28]. The BRI was developed to assess body shape independently of height. It is a better predictor of body fat and the percentage of visceral tissue compared with traditional indicators, such as the BMI and WC [29]. The discriminatory power of the BRI in predicting MetS was similar to that of the WC and only slightly lower than that of the WHtR [30]. It was higher than that of the BMI and ABSI in both sexes. In our previous paper we analyzed the usefulness of these indices to identify people with MetS [18]. In this study, we sought to validate whether the anthropometric indices previously recommended can also be applied to identify single MetS components to avoid a late diagnosis. Our aims were to assess the usefulness of the anthropometric indices and to determine optimal cut-off points for the identification of the MetS components in adults.

## Materials and methods

### Study design and participants

The base population of the study was 13,172 participants in the Polish-Norwegian Study (PONS). From this group, 844 individuals (6.4%) were excluded due to incomplete anthropometric measurements and/or biochemical parameters that were necessary to perform this analysis. Consequently, this study was carried out on 12,328 participants aged 55.7±5.4 years. The group included 4094 men. All the participants came from the Świętokrzyski region in Poland

and were Caucasian. The study design, recruitment of participants, and course of the study were described in detail in our previous publications [18, 31].

Consent to collect the data was given by the Ethics Committee from the Cancer Centre and Institute of Oncology in Warsaw, Poland (No. 69/2009/1/2011). Consents to carry out analyses of the data were given by the Committee on Bioethics at the Faculty of Health Sciences from Jan Kochanowski University in Kielce (No. 45/2016). Informed written consent was obtained from all participants enrolled in the study.

Anthropometric obesity indices, Blood Pressure and Serum Biochemical Parameters Weight, height, and WC were used to calculate obesity indices. All anthropometric measurements were conducted by trained nurses using standard protocols and techniques [32]. WC was measured in the horizontal plane midway between the lower rib edge and the upper iliac crest using a non-elastic metric measure. Height was measured using a stadiometer. Weight and %BF were measured with a body composition analyzer (Tanita SC 240MA). Blood pressure (BP) was measured on the right upper limb artery with an Omron pressure monitor (Model M3 Intellisense). The mean of two measurements was used for subsequent analysis. Serum TGs were measured with an enzymatic method using phosphoglycerol oxidase and determination of $H_2O_2$ (with peroxidase). HDL cholesterol was measured with a colorimetric non-precipitation method, using polyethylene glycol-modified enzymes. The glucose concentration was measured with an enzymatic method using hexokinase.

We considered the following obesity indices: WC, BMI, WHtR, %BF, CUN-BAE [19], BRI [29], and ABSI [26]. The following equations were used:

- BMI = weight (kg) / height $(m)^2$;

- WHtR = WC (cm) / height (cm);

- CUN-BAE was calculated using the following equation: %BF = $-$ 44.988 + (0.503 $\times$ age) + (10.689 $\times$ sex) + (3.172 $\times$ BMI) $-$ (0.026 $\times$ $BMI^2$) + (0.181 $\times$ BMI $\times$ sex) $-$ (0.02 $\times$ BMI $\times$ age) $-$ (0.005 $\times$ $BMI^2$ $\times$ sex) + (0.00021 $\times$ $BMI^2$ $\times$ age), where age is measured in years and sex was codified as 0 for men and 1 for women [19];

- BRI = 364.2–365.5 x $\sqrt{1 - \left[\frac{(WC/(2\pi))^2}{(0.5 \ x \ Height)^2}\right]}$ [29];

- ABSI = WC (m) / $[BMI^{2/3}(kg/m^2)$ height$^{1/2}$ (m)] [26].

## The definition of metabolic risk factors

Four MetS components were included in the analysis: elevated BP ($\geq$130 mmHg and/or diastolic blood pressure $\geq$85 mmHg or drug treatment for hypertension), high glucose concentration ($\geq$100 mg/dl or $\geq$5.5 mmol/L or diabetes treatment), high TG concentration ($\geq$150 mg/dL or $\geq$1.7 mmol/L or drug treatment for elevated triglycerides), and low HDL cholesterol. (<40 mg/dL or <1.0 mmol/L in men and <50 mg/dl or <1.3 mmol/L in women or drug treatment for low HDL cholesterol) [21].

## Statistical analysis

All statistical analyses were performed using the statistical package Statistica 13.3 (TIBCO SOFTWARE INC, Polish version, PL, Cracow). The participants were divided into two groups, according to the presence or lack of a given metabolic disorder. Additionally, the analyses were performed for at least one MetS component. Groups divided according to the MetS components were compared with the non-parametric Mann-Whitney U-test because of the non-

normal distribution of all quantitative variables. The analyses were done separately for both sexes.

We performed receiver-operating characteristic (ROC) curve analyses to determine the discriminatory power of the anthropometric indices as classifiers. Next, we calculated the areas under the curve (AUCs) and the 95% confidence intervals (CI) to compare the discriminatory power of each index. Sensitivity was defined as the percentage of true positive scores according to the criteria of individual MetS components. Specificity was defined as the proportion of scores identified incorrectly. The AUC is a measure of the precision of a given index in the differentiation between individuals with metabolic disorders and without them. It also characterizes the probability of assigning a patient to the correct group. For the AUC, 0.5 was adopted as the bottom border line. The indices with the biggest AUC were considered the best. Optimal cut-off points for all seven obesity indicators were determined with Youden's J statistic using the following equation: $J_{max.}$ = Sensitivity + Specificity– 1. The index values corresponding to the maximum value of Youden's J statistic were recognized as optimal cut-off points for these indices.

## Results

All anthropometric indices were significantly higher in men with abnormal metabolic parameters than in men with parameters within the normal range (according to the definition of MetS by the International Diabetes Federation [IDF]) [21] ($p < 0.001$, Table 1). Men who were not diagnosed with any of the metabolic disorders had the lowest average indices (BMI = 25.76 kg/m$^2$, WC = 91.78 cm, and %BF = 23.06%). Men with abnormal HDL cholesterol concentration had the highest index values (BMI = 29.68 kg/m$^2$, WC = 102.76 cm, and %BF = 29.05%).

Table 1. Baseline anthropometric and laboratory characteristics of men (N = 4094).

| MetS Components | | BMI [kg/m$^2$] X±SD Me±IQR | WC [cm] X±SD Me±IQR | WHtR X±SD Me±IQR | %BF X±SD Me±IQR | ABSI [m$^{11/6}$ · kg$^{-2/3}$] X±SD Me±IQR | BRI X±SD Me±IQR | CUN-BAE [%] X±SD Me±IQR |
|---|---|---|---|---|---|---|---|---|
| BP | normal n = 722 | 26.73±3.46 26.49±4.31 | 94.56±9.17 94.00±11.00 | 0.546±0.054 0.544±0.070 | 24.62±6.16 24.00±7.20 | 0.080±0.004 0.081±0.005 | 4.60±0.67 4.58±0.89 | 27.50±4.51 27.36±6.17 |
| | abnormal n = 3372 | 28.83±3.94 28.53±4.84 | 100.17±10.39 100.00±13.00 | 0.579±0.061 0.576±0.078 | 27.63±6.41 27.10±7.90 | 0.081±0.004 0.081±0.005 | 4.96±0.75 4.91±0.98 | 30.31±4.80 30.15±6.03 |
| Glucose | normal n = 2237 | 27.61±3.64 27.28±4.48 | 96.83±9.73 96.00±13.00 | 0.559±0.057 0.557±0.074 | 25.84±6.23 25.20±7.50 | 0.081±0.004 0.081±0.005 | 4.75±0.71 4.70±0.92 | 28.70±4.61 28.54±5.88 |
| | abnormal n = 1857 | 29.49±4.05 29.10±4.90 | 102.01±10.50 101.00±13.00 | 0.589±0.062 0.585±0.077 | 28.63±6.42 28.00±8.00 | 0.081±0.004 0.081±0.005 | 5.07±0.76 5.02±0.97 | 31.15±4.84 30.97±5.93 |
| TG | normal n = 2452 | 27.72±3.80 27.43±4.68 | 97.15±10.08 97.00±13.00 | 0.561±0.059 0.558±0.076 | 25.90±6.32 25.30±7.60 | 0.081±0.004 0.081±0.005 | 4.77±0.73 4.73±0.96 | 28.88±4.79 28.75±6.02 |
| | abnormal n = 1642 | 29.57±3.90 29.18±4.74 | 102.21±10.16 102.00±13.00 | 0.590±0.060 0.587±0.075 | 28.90±6.27 28.30±7.60 | 0.081±0.004 0.081±0.003 | 5.09±0.74 5.02±0.94 | 31.20±4.65 30.98±5.97 |
| HDL | normal n = 3424 | 28.72±6.39 27.89±4.87 | 98.48±10.17 8.00±12.00 | 0.568±0.060 0.566±0.077 | 26.72±6.39 26.10±7.90 | 0.081±0.004 0.081±0.005 | 4.84±0.73 4.80±0.96 | 29.48±4.83 29.36±6.19 |
| | abnormal n = 670 | 29.68±4.03 29.20±5.11 | 102.76±10.86 102.00±14.00 | 0.596±0.064 0.591±0.079 | 29.05±6.48 28.50±7.50 | 0.082±0.005 0.081±0.005 | 5.17±0.78 5.09±0.96 | 31.49±4.72 31.18±6.05 |
| At least one MetS component | No n = 1755 | 25.76±3.10 25.57±3.85 | 91.78±8.78 91.50±11.50 | 0.532±0.052 0.528±0.070 | 23.06±5.77 22.55±6.30 | 0.080±0.004 0.080±0.005 | 4.47±0.66 4.45±0.83 | 26.22±4.19 26.29±5.53 |
| | Yes n = 2339 | 28.71±3.92 28.41±4.87 | 99.85±10.28 99.00±13.00 | 0.577±0.061 0.573±0.077 | 27.90±6.40 26.90±7.90 | 0.081±0.004 0.081±0.005 | 4.93±0.74 4.88±0.95 | 30.14±4.80 30.01±6.07 |

X–arithmetic mean, SD–standard deviation, Me–median, IQR–interquartile range, BP–blood pressure, TG–triglycerides, HDL–HDL cholesterol, MetS–metabolic syndrome, WC–waist circumference, BMI–body mass index, WHtR–waist-to-height ratio, %BF–body fat percentage, ABSI–a body shape index, BRI–body roundness index, CUN-BAE–Clínica Universidad de Navarra-body adiposity estimator

**Table 2. Baseline anthropometric and laboratory characteristics of women (N = 8234).**

| MetS Components | | BMI [kg/m$^2$] X±SD Me±IQR | WC [cm] X±SD Me±IQR | WHtR X±SD Me±IQR | %BF X±SD Me±IQR | ABSI [m$^{11/6}$ · kg$^{-2/3}$] X±SD Me±IQR | BRI X±SD Me±IQR | CUN-BAE [%] X±SD Me±IQR |
|---|---|---|---|---|---|---|---|---|
| BP | normal n = 2470 | 25.86±3.83 25.38±4.95 | 82.90±9.87 82.00±13.00 | 0.517±0.064 0.512±0.085 | 33.18±6.43 33.70±8.20 | 0.076±0.005 0.075±0.006 | 4.74±0.84 4.68±1.11 | 38.15±4.79 37.92±6.57 |
| | abnormal n = 5764 | 28.90±5.11 28.28±6.65 | 90.36±12.02 90.00±16.00 | 0.567±0.078 0.561±0.104 | 37.21±6.47 37.70±8.40 | 0.075±0.005 0.075±0.006 | 5.36±1.02 5.28±1.34 | 41.83±5.53 41.69±7.67 |
| Glucose | normal n = 5982 | 27.12±4.51 26.52±5.79 | 85.90±10.95 85.00±15.00 | 0.537±0.071 0.531±0.095 | 35.01±6.59 35.50±8.60 | 0.077±0.005 0.077±0.006 | 4.99±0.93 4.91±1.24 | 39.72±5.29 39.46±7.26 |
| | abnormal n = 2252 | 30.30±5.35 29.74±6.97 | 94.03±12.36 93.00±17.00 | 0.591±0.080 0.587±0.107 | 38.63±6.32 39.30±8.00 | 0.075±0.005 0.075±0.005 | 5.66±1.05 5.59±1.38 | 43.38±5.46 43.36±7.56 |
| TG | normal n = 5650 | 27.27±4.74 26.55±6.09 | 86.11±11.41 85.00±15.00 | 0.538±0.074 0.531±0.010 | 35.03±6.78 35.50±9.00 | 0.075±0.006 0.075±0.040 | 5.00±0.97 4.90±1.28 | 39.84±5.48 39.46±7.63 |
| | abnormal n = 2584 | 29.55±5.09 28.87±6.59 | 92.54±11.81 91.00±16.00 | 0.582±0.076 0.575±0.105 | 38.13±6.05 38.40±7.65 | 0.077±0.006 0.077±0.004 | 5.55±1.00 5.46±1.33 | 42.66±5.29 42.44±7.27 |
| HDL | normal n = 6665 | 27.63±4.87 26.93±6.24 | 87.17±11.74 86.00±16.00 | 0.545±0.076 0.538±0.103 | 35.53±6.77 36.00±9.00 | 0.076±0.006 0.080±0.005 | 5.08±0.99 4.99±1.31 | 40.26±5.55 39.97±7.66 |
| | abnormal n = 1569 | 29.49±5.08 28.80±6.60 | 92.19±11.80 91.00±16.00 | 0.581±0.077 0.573±0.105 | 38.00±6.09 38.30±7.70 | 0.077±0.006 0.077±0.004 | 5.55±1.01 5.46±1.34 | 42.68±5.26 42.50±7.20 |
| At least one MetS component | No n = 4786 | 25.39±3.69 24.91±4.59 | 81.40±9.43 80.00±13.00 | 0.507±0.077 0.556±0.103 | 32.39±4.71 37.27±8.00 | 0.076±0.005 0.076±0.006 | 4.62±0.81 4.54±1.09 | 37.49±4.71 37.27±6.14 |
| | Yes n = 3448 | 28.66±5.03 28.04±6.49 | 89.87±11.87 89.00±15.00 | 0.563±0.077 0.556±0.103 | 36.94±6.48 37.50±8.40 | 0.075±0.005 0.074±0.006 | 5.31±1.01 5.23±1.33 | 41.56±5.48 41.43±7.58 |

X–arithmetic mean, SD–standard deviation, Me–median, IQR–interquartile range, BP–blood pressure, TG–triglycerides, HDL–HDL cholesterol, MetS–metabolic syndrome, WC–waist circumference, BMI–body mass index, WHtR–waist-to-height ratio, %BF–body fat percentage, ABSI–a body shape index, BRI–body roundness index, CUN-BAE–Clínica Universidad de Navarra-body adiposity estimator

Similarly, all obesity indices were significantly higher in women with abnormal metabolic parameters than in those with normal parameters (p<0.001; Table 2). Women who were not diagnosed with any of the metabolic disorders had the lowest indices (BMI = 25.39 kg/m$^2$, WC = 81.40 cm, %BF = 32.39%). Women with abnormal glucose concentration had the highest index values (BMI = 30.30 kg/m$^2$, WC = 94.03 cm, and %BF = 38.63).

In men, the largest AUC for elevated BP, was for the CUN-BAE (0.668) and the BMI (0.660). For abnormal glucose concentration, the largest AUC was for the CUN-BAE (0.649) and the WC (0.645) (Table 3). The largest AUC for TGs was for the WC (0.642) and the %BF (0.641). The CUN-BAE, BMI, and WC also had high discriminatory power for at least one MetS component, with an AUC of 0.734, 0.728, and 0.728, respectively.

In women, the largest AUCs for elevated BP were for the CUN-BAE (0.691) and the WHtR (0.688) (Table 4). For abnormal glucose and TG concentration, the largest AUC was for the WHtR (0.694 and 0.664, respectively). The WHtR, CUN-BAE, and WC had the highest discriminatory power for at least one MetS component (0.715, 0.714, and 0.712, respectively). In both men and women, the largest AUCs for HDL were for the WHtR (0.624 and 0.632, respectively) and BRI (0.622 and 0.634, respectively).

For the BMI, the optimal cut-off point for an early diagnosis of single metabolic disorders (MetS components) was 27.2 kg/m$^2$ for both sexes. This point was the lowest for BP for men and TG for women. The optimal cut-off point for WC was 94 cm for men (for at least 1 component of MetS) and 87 cm for women (for glucose). For other indices, optimal cut-off points for the identification of single metabolic disorders in men include WHtR = 0.549, %BF = 25.6, CUN-BAE = 28.76%, BRI = 4.612, and ABSI = 0.080. For women, the optimal cut-off points were WHtR = 0.532, %BF = 35.3, CUN-BAE = 39.09%, BRI = 4.910, and ABSI = 0.076. The discriminatory power of the ABSI was the lowest both in men and women (AUC<0.6).

**Table 3. Areas under the curve (AUCs) and cut-off points for anthropometric indices for the prediction of MetS components in men.**

| MetS Components | Indices | AUC | 95%CI | p | Sensitivity | Specificity | Youden index | Cut-off points |
|---|---|---|---|---|---|---|---|---|
| BP | BMI | 0.660 | 0.638–0.681 | 0.000 | 0.651 | 0.594 | 0.245 | 27.18 |
| | WC | 0.657 | 0.636–0.678 | 0.000 | 0.590 | 0.639 | 0.228 | 98.00 |
| | WHtR | 0.655 | 0.633–0.676 | 0.000 | 0.582 | 0.647 | 0.228 | 0.564 |
| | %BF | 0.646 | 0.624–0.668 | 0.000 | 0.575 | 0.659 | 0.234 | 26.10 |
| | ABSI | 0.542 | 0.519–0.565 | 0.000 | 0.346 | 0.715 | 0.061 | 0.083 |
| | BRI | 0.638 | 0.616–0.659 | 0.000 | 0.538 | 0.677 | 0.215 | 4.855 |
| | CUN-BAE | 0.668 | 0.647–0.960 | 0.000 | 0.625 | 0.630 | 0.255 | 28.76 |
| Glucose | BMI | 0.641 | 0.625–0.659 | 0.000 | 0.629 | 0.601 | 0.228 | 28.15 |
| | WC | 0.645 | 0.628–0.661 | 0.000 | 0.702 | 0.503 | 0.206 | 97.00 |
| | WHtR | 0.641 | 0.624–0.658 | 0.000 | 0.666 | 0.534 | 0.199 | 0.561 |
| | %BF | 0.630 | 0.613–0.647 | 0.000 | 0.674 | 0.527 | 0.202 | 25.60 |
| | ABSI | 0.545 | 0.527–0.563 | 0.000 | 0.653 | 0.368 | 0.051 | 0.080 |
| | BRI | 0.624 | 0.607–0.641 | 0.000 | 0.663 | 0.519 | 0.182 | 4.737 |
| | CUN-BAE | 0.649 | 0.633–0.666 | 0.000 | 0.613 | 0.620 | 0.233 | 29.89 |
| TG | BMI | 0.639 | 0.622–0.656 | 0.000 | 0.648 | 0.560 | 0.208 | 27.93 |
| | WC | 0.642 | 0.625–0.659 | 0.000 | 0.600 | 0.618 | 0.218 | 100.00 |
| | WHtR | 0.639 | 0.622–0.656 | 0.000 | 0.688 | 0.513 | 0.201 | 0.560 |
| | %BF | 0.641 | 0.624–0.658 | 0.000 | 0.690 | 0.533 | 0.223 | 25.70 |
| | ABSI | 0.543 | 0.525–0.561 | 0.000 | 0.599 | 0.469 | 0.068 | 0.081 |
| | BRI | 0.622 | 0.605–0.639 | 0.000 | 0.605 | 0.425 | 0.180 | 4.859 |
| | CUN-BAE | 0.639 | 0.622–0.656 | 0.000 | 0.683 | 0.525 | 0.208 | 29.04 |
| HDL | BMI | 0.607 | 0.584–0.629 | 0.000 | 0.649 | 0.519 | 0.168 | 28.08 |
| | WC | 0.614 | 0.59–0.637 | 0.000 | 0.613 | 0.558 | 0.172 | 100.00 |
| | WHtR | 0.624 | 0.601–0.647 | 0.000 | 0.448 | 0.735 | 0.183 | 0.601 |
| | %BF | 0.608 | 0.586–0.631 | 0.000 | 0.693 | 0.483 | 0.175 | 25.90 |
| | ABSI | 0.556 | 0.532–0.580 | 0.000 | 0.357 | 0.676 | 0.073 | 0.083 |
| | BRI | 0.622 | 0.599–0.645 | 0.000 | 0.685 | 0.499 | 0.184 | 4.803 |
| | CUN-BAE | 0.619 | 0.596–0.641 | 0.000 | 0.715 | 0.473 | 0.187 | 29.04 |
| At least one MetS component | BMI | 0.728 | 0.702–0.755 | 0.000 | 0.586 | 0.776 | 0.362 | 27.65 |
| | WC | 0.728 | 0.701–0.755 | 0.000 | 0.740 | 0.615 | 0.355 | 94.00 |
| | WHtR | 0.715 | 0.687–0.742 | 0.000 | 0.617 | 0.656 | 0.327 | 0.549 |
| | %BF | 0.717 | 0.689–0.744 | 0.000 | 0.575 | 0.774 | 0.348 | 25.90 |
| | ABSI | 0.562 | 0.530–0.593 | 0.000 | 0.566 | 0.526 | 0.092 | 0.081 |
| | BRI | 0.682 | 0.653–0.710 | 0.000 | 0.658 | 0.618 | 0.276 | 4.612 |
| | CUN-BAE | 0.734 | 0.708–0.761 | 0.000 | 0.611 | 0.759 | 0.370 | 28.76 |

BP–blood pressure, TG–triglycerides, HDL–HDL cholesterol, MetS–metabolic syndrome, WC–waist circumference, BMI–body mass index, WHtR–waist-to-height ratio, %BF–body fat percentage, ABSI–a body shape index, BRI–body roundness index, CUN-BAE–Clínica Universidad de Navarra-body adiposity estimator

## Discussion

The usefulness of anthropometric indices for predicting single metabolic disorders that are MetS components was different depending on sex and the type of metabolic abnormality. In men, the CUN-BAE had the largest AUC, followed by the BMI and WC. In women, the largest AUCs were those of the WHtR, CUN-BAE, and WC. However, the differences between the AUCs of most indices were small. Therefore, the predictive power of most of the indices is similar. In men, indices with a similar predictive power to those with the largest AUCs include the %BF and WHtR. In women, they include the BRI, BMI, and %BF. One reason for the

**Table 4. Areas under the curve (AUCs) and cut-off points for anthropometric indices for the prediction of MetS components in women.**

| MetS Components | Indices | AUC | 95%CI | p | Sensitivity | Specificity | Youden index | Cut-off points |
|---|---|---|---|---|---|---|---|---|
| BP | BMI | 0.681 | 0.669–0.693 | 0.001 | 0.488 | 0.783 | 0.271 | 28.44 |
| | WC | 0.684 | 0.672–0.696 | 0.001 | 0.564 | 0.712 | 0.275 | 88.00 |
| | WHtR | 0.688 | 0.676–0.700 | 0.001 | 0.631 | 0.649 | 0.279 | 0.536 |
| | %BF | 0.679 | 0.666–0.691 | 0.001 | 0.567 | 0.704 | 0.271 | 36.80 |
| | ABSI | 0.575 | 0.562–0.589 | 0.001 | 0.383 | 0.727 | 0.111 | 0.078 |
| | BRI | 0.681 | 0.669–0.693 | 0.001 | 0.635 | 0.366 | 0.269 | 4.931 |
| | CUN-BAE | 0.691 | 0.679–0.703 | 0.001 | 0.587 | 0.701 | 0.287 | 40.52 |
| Glucose | BMI | 0.681 | 0.668–0.694 | 0.001 | 0.597 | 0.679 | 0.276 | 28.60 |
| | WC | 0.691 | 0.678–0.704 | 0.001 | 0.716 | 0.559 | 0.276 | 87.00 |
| | WHtR | 0.694 | 0.681–0.707 | 0.001 | 0.612 | 0.675 | 0.287 | 0.564 |
| | %BF | 0.662 | 0.649–0.675 | 0.001 | 0.591 | 0.660 | 0.250 | 38.00 |
| | ABSI | 0.595 | 0.581–0.608 | 0.001 | 0.624 | 0.517 | 0.141 | 0.076 |
| | BRI | 0.686 | 0.673–0.699 | 0.001 | 0.578 | 0.699 | 0.278 | 5.400 |
| | CUN-BAE | 0.688 | 0.675–0.700 | 0.001 | 0.635 | 0.652 | 0.286 | 41.56 |
| TG | BMI | 0.636 | 0.623–0.649 | 0.001 | 0.650 | 0.558 | 0.208 | 27.20 |
| | WC | 0.657 | 0.644–0.669 | 0.001 | 0.641 | 0.592 | 0.233 | 88.00 |
| | WHtR | 0.664 | 0.651–0.676 | 0.001 | 0.676 | 0.562 | 0.238 | 0.543 |
| | %BF | 0.638 | 0.626–0.651 | 0.001 | 0.671 | 0.542 | 0.213 | 36.14 |
| | ABSI | 0.605 | 0.592–0.618 | 0.001 | 0.631 | 0.528 | 0.159 | 0.076 |
| | BRI | 0.661 | 0.648–0.673 | 0.001 | 0.731 | 0.504 | 0.235 | 4.910 |
| | CUN-BAE | 0.647 | 0.634–0.659 | 0.001 | 0.747 | 0.476 | 0.223 | 39.09 |
| HDL | BMI | 0.610 | 0.595–0.625 | 0.001 | 0.639 | 0.535 | 0.174 | 27.32 |
| | WC | 0.622 | 0.607–0.636 | 0.001 | 0.625 | 0.553 | 0.178 | 88.00 |
| | WHtR | 0.632 | 0.617–0.647 | 0.001 | 0.584 | 0.608 | 0.193 | 55.87 |
| | %BF | 0.607 | 0.593–0.622 | 0.001 | 0.713 | 0.454 | 0.167 | 35.30 |
| | ABSI | 0.578 | 0.563–0.594 | 0.001 | 0.617 | 0.500 | 0.117 | 0.076 |
| | BRI | 0.634 | 0.619–0.649 | 0.001 | 0.652 | 0.540 | 0.192 | 5.082 |
| | CUN-BAE | 0.625 | 0.610–0.640 | 0.001 | 0.652 | 0.544 | 0.196 | 40.61 |
| At least one MetS component | BMI | 0.702 | 0.688–0.715 | 0.001 | 0.555 | 0.756 | 0.311 | 27.41 |
| | WC | 0.712 | 0.699–0.726 | 0.001 | 0.547 | 0.771 | 0.317 | 88.00 |
| | WHtR | 0.715 | 0.702–0.728 | 0.001 | 0.634 | 0.689 | 0.322 | 0.532 |
| | %BF | 0.701 | 0.688–0.714 | 0.001 | 0.589 | 0.723 | 0.313 | 36.14 |
| | ABSI | 0.603 | 0.588–0.618 | 0.001 | 0.553 | 0.597 | 0.149 | 0.076 |
| | BRI | 0.706 | 0.692–0.719 | 0.001 | 0.618 | 0.693 | 0.310 | 4.934 |
| | CUN-BAE | 0.714 | 0.701–0.727 | 0.001 | 0.570 | 0.759 | 0.329 | 40.85 |

BP–blood pressure, TG–triglycerides, HDL–HDL cholesterol, MetS–metabolic syndrome, WC–waist circumference, BMI–body mass index, WHtR–waist-to-height ratio, %BF–body fat percentage, ABSI–a body shape index, BRI–body roundness index, CUN-BAE–Clínica Universidad de Navarra-body adiposity estimator

similarity in the predictive power between the indices may be that they are correlated to a large extent [18]. The ABSI had the lowest discriminatory power for predicting metabolic disorders in both sexes (AUC<0.6).

Other authors have reached diverse conclusions concerning the predominance of particular anthropometric indices over other indicators in diagnosing obesity and metabolic disorders. Our results are similar to those obtained in the population of Jordan [33]. Khader et al. recommend the use of the WHtR for the diagnosis of metabolic disorders defined according to the IDF. However, none of the anthropometric indices included in their analysis (i.e. BMI, WC,

WHR, and WHtR) was significantly better than the others in identifying most of these disorders. Similarly, in studies conducted in the Spanish (Caucasian) population, all obesity indices, except for the ABSI, had similar discriminatory power in the prediction of MetS [34]. When both sexes were analyzed separately, the BMI had the largest AUC in men and the WHtR and BRI had the largest AUCs in women. Similar to the results of our analysis in men, Davila-Batista et al. concluded that the CUN-BAE was the best index for the identification of individuals with hypertension, diabetes, and MetS [35]. Corbatón Anchuelo et al. emphasize the advantages of using the WC and WHtR in the identification of cardiometabolic disorders in women [16]. In the Chinese population, the best predictors of cardiometabolic disorders in men and women were the BRI and WHtR [36]. Moreover, these indices were the best predictors of elevated BP and the presence of at least one (any) metabolic disorder in men. Also in men, the BMI and WC were the best indices for the identification of dyslipidemia and MetS. The ABSI had the lowest discriminatory power, in agreement with our findings. The results depend to a large extent on the ethnicity [37, 38], sex [36, 37], and age of the participants [39], and also on the indices selected for analysis.

According to the National Cholesterol Education Program, Adult Treatment Panel III (NCEP ATP III), the cut-off point for abdominal obesity is 102 cm for men and 88 cm for women [21]. These values correspond to a BMI of about 30 kg/m$^2$. However, according to the IDF criteria, abdominal obesity in Europeans should be recognized when the WC is 94 cm for men and 80 cm for women. In a study on the Saudi population, the optimal cut-off point for the WC for the identification of at least 2 components of MetS was 92 cm for men and 87 cm for women [40]. In a study conducted in Jordan, the cut-off point for the WC for the identification of single metabolic disorders fluctuated between 88.5 and 91.8 cm in men. Analogically, the cut-off point in women ranged from 84.5 to 88.5 cm [33]. From these studies and our own observations, we can conclude that optimal cut-off point for the WC is 94 cm for men and 87 cm for women. Therefore, the IDF criteria are more useful for the early identification of metabolic disorders, especially in men. The optimal cut-off point for the WHtR for the identification of MetS was 0.549 for men and 0.532 for women and was only slightly higher than the cut-off point for abdominal obesity ($\geq$0.5) [41]. For the CUN-BAE, the cut-off point for the diagnosis of single metabolic disorders was 28.76% for men and 39.09% for women. In the MARK Study, Gomez-Marcos et al. obtained slightly higher cut-off points for the identification of individuals with MetS, according to the criteria of NCEP ATP III (31.22% for men and 41.95% for women) [34]. An optimal cut-off point for the BMI for the identification of metabolic disorders is 27.2 kg/m$^2$ for both sexes. Similar values for MetS were obtained in a study conducted in Israel (27 kg/m$^2$) [42]. Compared with our results, the BMI cut-off values for individual metabolic disorders obtained in studies conducted in Jordan, were slightly lower in men (26.2–27.2 kg/m$^2$), but higher in women (27.2–30.0 kg/m$^2$) [33].

Considering there is a high probability that the indices will have considerable discriminatory power to identify individuals with single metabolic disorders, we recommend using the WHtR and the WC because they are the simplest to measure and interpret. For the WC, the IDF criteria should be preferred, especially in men. We also confirm the usefulness of the CUN-BAE. Despite being based on the BMI, it has the advantage of allowing for the age and sex of participants. In their long-term study on a population of European descent, Vinknes et al. found that the CUN-BAE was more strongly correlated with the risk of cardiovascular diseases and diabetes than the BMI [43]. The cut-off points for the identification of single metabolic disorders, were slightly lower than the optimal points for the identification $\geq$2 and $\geq$3 MetS components in the same population [18]. Our results allow us to conclude that these anthropometric indices can be used to identify individuals with single metabolic disorders.

However, they are more useful for the identification of individuals with at least 2 MetS components [18].

## Limitations

The presence of MetS or its components in individuals with a normal body mass (metabolically obese normal weight—MONW) indicates that not all cases of metabolic disorders are characterized by high anthropometric indices. MetS can be related not only to excessive adipose tissue, but also to the location of this tissue and changes in its functions. Anatomical and/or functional changes in adipose tissue, promoted by a positive energy balance, are responsible for the so-called adiposopathy ("sick fat") in genetically susceptible people. Hormonal and immunological reactions caused by adiposopathy can make metabolic disorders more severe, for example by causing dyslipidemia and elevated BP [44]. Moreover, genetic variation and epigenetic factors play an important role in the pathogenesis of MetS [45]. Eating habits and physical activity are the main environmental factors conditioning the expression of genes involved in the occurrence of MetS [46, 47]. Diets that are rich in fats, especially in saturated fatty acids, with a high glycemic index, and a low fiber content can increase the risk of a MetS. Conversely, diets characterized by a low consumption of sugar, sweets, refined grains, processed meat, and high consumption of fish, legumes, nuts, whole grains, and phytochemical-rich foods decrease the risk of metabolic disorders [48–50]. Interventions involving physical activity also positively influence each of the MetS components [46, 47]. Several genes associated with the ability of skeletal muscles to use lipids have been identified, which helps explain how physical activity affects the concentration of lipids in the blood [51].

## Conclusions

We recommend the following indices of nutritional status for the identification of the MetS components: CUN-BAE>BMI = WC in men and WHtR>CUN-BAE>WC in women. Except for the ABSI, the diagnostic value of the other indices we analyzed was very similar. Prospective studies are needed to identify those indices in which changes in value predict the development of metabolic disorders best.

In the diagnosis of metabolic disorders, the cut-off point for the WC should be considered in accordance with the IDF rather than the NCEP ATP III, especially for men.

## Supporting information

**S1 File. Data.**
(PDF)

## Acknowledgments

We are grateful to the members of the PONS project team for their contribution to the study and data sharing.

## Author Contributions

**Conceptualization:** Stanisław Głuszek, Elzbieta Ciesla, Martyna Głuszek-Osuch, Dorota Kozieł, Wojciech Kiebzak, Łukasz Wypchło, Edyta Suliga.

**Formal analysis:** Elzbieta Ciesla, Łukasz Wypchło, Edyta Suliga.

**Funding acquisition:** Stanisław Głuszek, Dorota Kozieł, Edyta Suliga.

**Methodology:** Stanisław Głuszek, Elzbieta Ciesla, Edyta Suliga.

**Supervision:** Stanisław Głuszek, Wojciech Kiebzak, Edyta Suliga.

**Writing – original draft:** Martyna Głuszek-Osuch, Łukasz Wypchło, Edyta Suliga.

**Writing – review & editing:** Elzbieta Ciesla, Martyna Głuszek-Osuch, Dorota Kozieł, Wojciech Kiebzak.

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
