## [Decision Letter · Decision Letter 0]

14 Apr 2020

PONE-D-20-07763

Anthropometric indices and cut-off points in the diagnostics of the metabolic disorders

PLOS ONE

Dear Professor Suliga,

Thank you for submitting your manuscript to PLOS ONE. After careful consideration, we feel that it has merit but does not fully meet PLOS ONE’s publication criteria as it currently stands. Therefore, we invite you to submit a revised version of the manuscript that addresses the points raised during the review process.

We would appreciate receiving your revised manuscript by May 29 2020 11:59PM. To enhance the reproducibility of your results, we recommend that if applicable you deposit your laboratory protocols in protocols.io, where a protocol can be assigned its own identifier (DOI) such that it can be cited independently in the future. For instructions see: http://journals.plos.org/plosone/s/submission-guidelines#loc-laboratory-protocols

We look forward to receiving your revised manuscript.

Kind regards,

Kiyoshi Sanada, PhD

Academic Editor

PLOS ONE

Journal Requirements:

2. Please note that outmoded terms and potentially stigmatizing labels should be changed to more current, acceptable terminology. Examples: “Caucasian” should be changed to “white” or “of [Western] European descent” (as appropriate); “cancer victims” should be changed to “patients with cancer.” https://journals.plos.org/plosone/s/submission-guidelines#loc-human-subjects-research.

4. Your ethics statement must appear in the Methods section of your manuscript. If your ethics statement is written in any section besides the Methods, please move it to the Methods section and delete it from any other section. Please also ensure that your ethics statement is included in your manuscript, as the ethics section of your online submission will not be published alongside your manuscript.

Reviewers' comments:

Reviewer's Responses to Questions

**Comments to the Author**

1. Is the manuscript technically sound, and do the data support the conclusions?

Reviewer #1: Partly

Reviewer #2: Yes

2. Has the statistical analysis been performed appropriately and rigorously? 

Reviewer #1: Yes

Reviewer #2: Yes

3. Have the authors made all data underlying the findings in their manuscript fully available?

Reviewer #1: Yes

Reviewer #2: Yes

4. Is the manuscript presented in an intelligible fashion and written in standard English?

Reviewer #1: No

Reviewer #2: No

5. Review Comments to the Author

Reviewer #1: The authors obtained cut-off points of anthropometric indices in the diagnostics of the metabolic disorders. I think these cut-off points are important. But, there are several problems. I have the following comments.

Seven indices (WC, BMI, WHtR, %BF, CUN-BAE, BRI, and ABSI) were taken into consideration in this study. Why did you chose these indices? Please describe the reason in Introduction. Also, please explain the relationship between these indices and metabolic disorders based on previous studies, simply.

Is the base population of this study 13,172 (Suliga et al. 2019)?

The authors described “This analysis was conducted on the data of 12,328 participants” in this manuscript. How were these 12,328 participants selected? Please describe the process. Please show a flowchart, if possible.

Please describe how each index was calculated and how components of metabolic syndrome were evaluated.

Who measured the waist circumference? I think the accuracy of the waist circumference measurement is important. Please show reproducibility, if possible.

There are some unexplained abbreviations in the manuscript (e.g., LDL, IDF, NCEP ATP III, MONW). Please modify.

The number of normal and abnormal participants in each variable was not shown. Table 1 and 2 are too busy. Why are both the mean and median shown in table 1 and 2? Please modify.

The value in the text is different from the value in the table. Also, there are “men/women” and “male/female” in the manuscript. Please check.

Reviewer #2: Major comments

1. There are various types of metabolic syndrome risk. More should be mentioned about which factors are associated with what metabolic risk in the Introduction. You should then add any issues or hypotheses. Moreover What is the difference between the selected factors? Please mention as allometric scaling.

2. Throughout the text, data should be considered with significant figures. For example, age does not require a second decimal place.

3. Subjects include persons with patients. Please clarify the exclusion criteria for subjects.

4. Regarding the criteria for metabolic syndrome, do you need to consider separately the classification based on the reference value and the drug taker? The conditions differ for those who only meet the criteria and those who take medication.

5. Title the statistical analysis and summarize it. There is no description of Me ± IQR. Do you need it?

6. The purpose of this study is to compare each factor, so Table 1 needs ANCOVA. At least covariates need age, sex, and smoking status.

7. Please state the conclusion concisely.

6. PLOS authors have the option to publish the peer review history of their article (what does this mean?). If published, this will include your full peer review and any attached files.

Reviewer #1: No

Reviewer #2: No

---

## [Author Response · Author response to Decision Letter 0]

18 May 2020

Reviewer #1: 

Seven indices (WC, BMI, WHtR, %BF, CUN-BAE, BRI, and ABSI) were taken into consideration in this study. Why did you chose these indices? Please describe the reason in Introduction. Also, please explain the relationship between these indices and metabolic disorders based on previous studies, simply.

The Introduction has been supplemented with information on indices, as suggested by the reviewer.

Is the base population of this study 13,172 (Suliga et al. 2019)?

The authors described “This analysis was conducted on the data of 12,328 participants” in this manuscript. How were these 12,328 participants selected? Please describe the process. Please show a flowchart, if possible.

The base population of this study was 13,172. From this group, 844 individuals (6.4%) were rejected due to a lack of complete anthropometric measurements and / or biochemical parameters that were necessary to perform this analysis. Consequently, the study was carried out on 12,328 participants.

Please describe how each index was calculated and how components of metabolic syndrome were evaluated. Who measured the waist circumference? I think the accuracy of the waist circumference measurement is important. Please show reproducibility, if possible.

In the text of the study, we have included formulas for calculating the indicators. Information on anthropometric measurements and metabolic factor analyzes has also been provided.

There are some unexplained abbreviations in the manuscript (e.g., LDL, IDF, NCEP ATP III, MONW). Please modify.

Abbreviations have been explained.

The number of normal and abnormal participants in each variable was not shown. Table 1 and 2 are too busy. Why are both the mean and median shown in table 1 and 2? Please modify.

The numbers of participants with normal and abnormal metabolic parameters have been provided. Tables 1 and 2 have been corrected to remove unnecessary information.

The value in the text is different from the value in the table. Also, there are “men/women” and “male/female” in the manuscript. Please check.

Male / female "has been corrected to" men / women ". The differences in values in the tables and in the text have been explained. The values given in the text have been checked again with the data in the tables. We suppose that the incompatibilities noted by the reviewer may result from the fact that as optimal cut-off points, we ultimately adopted those that were the lowest and were not always points for "at least 1 component of MetS". The result depended, to a large extent, on the risk factor which was most common in the participants of the study. An appropriate explanation has been included in the text.

Reviewer #2: Major comments

1. There are various types of metabolic syndrome risk. More should be mentioned about which factors are associated with what metabolic risk in the Introduction. You should then add any issues or hypotheses. Moreover What is the difference between the selected factors? Please mention as allometric scaling.

We agree that there are various types of metabolic syndrome risks. However, due to the extensive amount of information, we decided to present only those factors that were the subject of analysis in our work. We have only supplemented them partially, because our introduction is already relatively long.

Our analysis focused on the search for cut-off points for 7 anthropometric indicators and their possible future use in the approach to metabolic disorders. Therefore, in our opinion, it was not advisable to include an allometric approach in this situation.

2. Throughout the text, data should be considered with significant figures. For example, age does not require a second decimal place.

The data were corrected as suggested by the reviewer.

3. Subjects include persons with patients. Please clarify the exclusion criteria for subjects.

Our project was an epidemiological study aimed at determining the prevalence of selected non-communicable diseases and their risk factors. Participants volunteered for the study. Therefore, the study group included both healthy people and those who had already begun pharmacological treatment for various metabolic disorders.

4. Regarding the criteria for metabolic syndrome, do you need to consider separately the classification based on the reference value and the drug taker? The conditions differ for those who only meet the criteria and those who take medication.

According to the definitions of both NCEP ATP III and IDF, not only abnormal values of a given parameter (e.g. blood pressure) are considered as components of the metabolic syndrome, but also those taking medications for this reason are taken into consideration. Therefore, we conducted analyzes for all persons meeting these criteria, guided by the above definitions.

5. Title the statistical analysis and summarize it. There is no description of Me ± IQR. Do you need it?

The title of the subsection "Statistical Analysis" was on page 4, line 114. The description of statistical analyzes in our opinion does not differ from the applicable standards. Me ± IQR was removed as suggested by the reviewer.

6. The purpose of this study is to compare each factor, so Table 1 needs ANCOVA. At least covariates need age, sex, and smoking status.

The purpose of this study was the evaluation of the usefulness of selected indices and the determination of optimal cut-off points for the identification of metabolic disorders being the components of metabolic syndrome. Therefore, we did not create models estimating the impact of confounding variables on the risk of metabolic disorders. In order to determine the discriminatory power of anthropometric indices as classifiers, receiver-operating characteristic (ROC) analyzes were performed. Next, in order to compare the discriminatory power of each index, areas under the curve ROC (AUC) were calculated. The AUC comprises a measure of precision of a given index in differentiation between individuals with metabolic disorders or without them , as well as characterizing the probability of a correct classification of a participant to a correct group. To our knowledge, this is the standard way of determining the discriminatory power of anthropometric indices and determining optimal cut-off points. Tables 1 and 2 present only the basic characteristics of the study group. The main purpose of using covariance analysis (ANCOVA) is to check to what extent the average of the analyzed parameters change under the influence of control variables and studied interactions. The use of covariance analysis in this situation would rather change the approach to the research problem raised in this work, creating a completely new problem that can be the subject of different analysis and the purpose of another work.

7. Please state the conclusion concisely.

Conclusions were corrected as suggested by the reviewer

---

## [Decision Letter · Decision Letter 1]

1 Jun 2020

PONE-D-20-07763R1

Anthropometric indices and cut-off points in the diagnosis of metabolic disorders

PLOS ONE

Dear Dr. Suliga,

Thank you for submitting your manuscript to PLOS ONE. After careful consideration, we feel that it has merit but does not fully meet PLOS ONE’s publication criteria as it currently stands. Therefore, we invite you to submit a revised version of the manuscript that addresses the points raised during the review process.

We look forward to receiving your revised manuscript.

Kind regards,

Kiyoshi Sanada, PhD

Academic Editor

PLOS ONE

Reviewers' comments:

Reviewer's Responses to Questions

**Comments to the Author**

1. If the authors have adequately addressed your comments raised in a previous round of review and you feel that this manuscript is now acceptable for publication, you may indicate that here to bypass the “Comments to the Author” section, enter your conflict of interest statement in the “Confidential to Editor” section, and submit your "Accept" recommendation.

Reviewer #1: (No Response)

Reviewer #2: All comments have been addressed

2. Is the manuscript technically sound, and do the data support the conclusions?

Reviewer #1: Yes

Reviewer #2: Yes

3. Has the statistical analysis been performed appropriately and rigorously? 

Reviewer #1: Yes

Reviewer #2: Yes

4. Have the authors made all data underlying the findings in their manuscript fully available?

Reviewer #1: Yes

Reviewer #2: Yes

5. Is the manuscript presented in an intelligible fashion and written in standard English?

Reviewer #1: Yes

Reviewer #2: Yes

6. Review Comments to the Author

Reviewer #1: This manuscript has been properly modified. But I have some minor comments.

1) In response to previous review, the authors described " The base population of this study was 13,172. From this group, 844 individuals (6.4%) were rejected due to a lack of complete anthropometric measurements and/or biochemical parameters that were necessary to perform this analysis". This information should be added to Study Design and Participants.

2) Page 8 Line 210: “norm”. “normal”? Please check.

3) The authors described “The largest AUC for TGs was for the WC (0.642) and the %BF (0.641)” in Page 9 Line 227. I think the position of this sentence should be changed. The position before " The CUN-BAE, BMI, and WC also had high discriminatory power for at least one MetS component, with an AUC of 0.734, 0.728, and 0.728, respectively" is better.

4) The authors described “In both men and women, the largest AUCs were for the WHTtR (0.624 and 0.632, respectively) and BRI (0.622 and 0.634, respectively)” in Page 10 Line 240.

Only this result (WHtR) was explained for both men and women. Please describe in each paragraph.

In addition, please modify “WHTtR”.

5) In the text, the ABSI optimal cut-off point for men is 0.079 (Page 11 Line 255). But it is 0.080 in the Table3. Which is correct? Please check.

6) The description of WHtR value is not unified.

Text: For other indices, optimal cut-off points for the identification of single metabolic disorders in men include WHtR=0.549… etc.

Table3: 56.36 (BP) … etc.

Table4: 0.536 (BP) … etc.

Please unify.

Reviewer #2: (No Response)

7. PLOS authors have the option to publish the peer review history of their article (what does this mean?). If published, this will include your full peer review and any attached files.

Reviewer #1: No

Reviewer #2: No

---

## [Author Response · Author response to Decision Letter 1]

8 Jun 2020

Response to Reviewers

1) In response to previous review, the authors described " The base population of this study was 13,172. From this group, 844 individuals (6.4%) were rejected due to a lack of complete anthropometric measurements and/or biochemical parameters that were necessary to perform this analysis". This information should be added to Study Design and Participants.

This information has been added as suggested by the reviewer.

2) Page 8 Line 210: “norm”. “normal”? Please check.

The sentence has been corrected.

3) The authors described “The largest AUC for TGs was for the WC (0.642) and the %BF (0.641)” in Page 9 Line 227. I think the position of this sentence should be changed. The position before " The CUN-BAE, BMI, and WC also had high discriminatory power for at least one MetS component, with an AUC of 0.734, 0.728, and 0.728, respectively" is better.

The position of this sentence is changed.

4) The authors described “In both men and women, the largest AUCs were for the WHTtR (0.624 and 0.632, respectively) and BRI (0.622 and 0.634, respectively)” in Page 10 Line 240.

Only this result (WHtR) was explained for both men and women. Please describe in each paragraph.

In addition, please modify “WHTtR”.

While editing the text, information about the HDL-cholesterol was mistakenly removed from the above sentence. That is why the meaning changed. It has been corrected. 

“WHTtR” is modified. 

5) In the text, the ABSI optimal cut-off point for men is 0.079 (Page 11 Line 255). But it is 0.080 in the Table3. Which is correct? Please check.

 The ABSI optimal cut-off point for men = 0.080 is correct.

6) The description of WHtR value is not unified.

Text: For other indices, optimal cut-off points for the identification of single metabolic disorders in men include WHtR=0.549… etc.

Table3: 56.36 (BP) … etc.

Table4: 0.536 (BP) … etc.

Please unify.

The description of WHtR value is unified.

---

## [Editor Report · Decision Letter 2]

10 Jun 2020

Anthropometric indices and cut-off points in the diagnosis of metabolic disorders

PONE-D-20-07763R2

Dear Dr. Suliga,

We’re pleased to inform you that your manuscript has been judged scientifically suitable for publication and will be formally accepted for publication once it meets all outstanding technical requirements.

Kind regards,

Kiyoshi Sanada, PhD

Academic Editor

PLOS ONE
---

## [Editor Report · Acceptance letter]

11 Jun 2020

PONE-D-20-07763R2 

Anthropometric indices and cut-off points in the diagnosis of metabolic disorders 

Dear Dr. Suliga:

I'm pleased to inform you that your manuscript has been deemed suitable for publication in PLOS ONE. Congratulations! Your manuscript is now with our production department. 

Kind regards, 

on behalf of

Dr. Kiyoshi Sanada 

Academic Editor

PLOS ONE